# Radiomics in PI-RADS 3 Multiparametric MRI for Prostate Cancer Identification: Literature Models Re-Implementation and Proposal of a Clinical–Radiological Model

**DOI:** 10.3390/jcm11216304

**Published:** 2022-10-26

**Authors:** Andrea Corsi, Elisabetta De Bernardi, Pietro Andrea Bonaffini, Paolo Niccolò Franco, Dario Nicoletta, Roberto Simonini, Davide Ippolito, Giovanna Perugini, Mariaelena Occhipinti, Luigi Filippo Da Pozzo, Marco Roscigno, Sandro Sironi

**Affiliations:** 1Department of Radiology, ASST Papa Giovanni XXIII, Piazza OMS 1, 24127 Bergamo, Italy; 2School of Medicine, University of Milano-Bicocca, Piazza dell’Ateneo Nuovo 1, 20126 Milan, Italy; 3Medicine and Surgery Department, University of Milano-Bicocca, Via Cadore 48, 20900 Monza, Italy; 4Interdepartmental Research Centre Bicocca Bioinformatics Biostatistics and Bioimaging Centre-B4, University of Milano-Bicocca, Via Follereau 3, 20854 Vedano al Lambro, Italy; 5Department of Radiology, San Gerardo Hospital, Via G. B. Pergolesi 33, 20900 Monza, Italy; 6Radiomics, Boulevard Patience et Beaujonc 3, 4000 Liège, Belgium; 7Department of Urology, ASST Papa Giovanni XXIII, Piazza OMS 1, 24127 Bergamo, Italy

**Keywords:** PI-RADS 3, prostate cancer, MRI, radiomics, texture analysis

## Abstract

PI-RADS 3 prostate lesions clinical management is still debated, with high variability among different centers. Identifying clinically significant tumors among PI-RADS 3 is crucial. Radiomics applied to multiparametric MR (mpMR) seems promising. Nevertheless, reproducibility assessment by external validation is required. We retrospectively included all patients with at least one PI-RADS 3 lesion (PI-RADS v2.1) detected on a 3T prostate MRI scan at our Institution (June 2016–March 2021). An MRI-targeted biopsy was used as ground truth. We assessed reproducible mpMRI radiomic features found in the literature. Then, we proposed a new model combining PSA density and two radiomic features (texture regularity (T2) and size zone heterogeneity (ADC)). All models were trained/assessed through 100-repetitions 5-fold cross-validation. Eighty patients were included (26 with GS ≥ 7). In total, 9/20 T2 features (Hector’s model) and 1 T2 feature (Jin’s model) significantly correlated to biopsy on our dataset. PSA density alone predicted clinically significant tumors (sensitivity: 66%; specificity: 71%). Our model obtained a sensitivity of 80% and a specificity of 76%. Standard-compliant works with detailed methodologies achieve comparable radiomic feature sets. Therefore, efforts to facilitate reproducibility are needed, while complex models and imaging protocols seem not, since our model combining PSA density and two radiomic features from routinely performed sequences appeared to differentiate clinically significant cancers.

## 1. Introduction

Prostate cancer (PC) is the second leading tumor in the male population worldwide [1]. Multiparametric magnetic resonance imaging (mpMRI) is the gold standard for prostate cancer imaging nowadays, proven to be helpful in early diagnosis, being employed in the evaluation of prostate gland lesions, local T-staging or recurrence, and in the assessment of pelvic lymph nodes involvement [2] along with Prostate Imaging Reporting and Data System version 2.1 (PI-RADS v2.1) guidelines [3]. Many studies have demonstrated a high correlation between PI-RADS and the Gleason score (GS) of prostate lesions [4,5,6,7]. However, while PI-RADS 4/5 are considered highly suspicious for neoplasia, the presence of clinically significant cancer in PI-RADS 3 lesions is equivocal (16–21% reported prevalence) [8,9]. Consequently, there is no consensus on the clinical management of PI-RADS 3 lesions, with high variability in protocols used in different centers [10]. A prostate biopsy is mandatory for diagnosis, but it is associated with possible complications (prostatitis, urinary tract infections, and sepsis), which may lead to hospitalization and, in the worst cases, even death. Therefore, it is crucial to timely identify clinically significant tumors (i.e., lesions with a Gleason Score (GS) ≥ 7, according to current literature [11]) among PI-RADS 3 lesions [12,13].

Some single-center studies in the literature have tried to exploit mpMRI radiomic analysis to identify clinically significant prostate cancer (csPCa) with promising results [14,15,16,17,18,19]. However, each center found its own radiomic features pool, likely due to high variability in center-specific population features, gold-standard definition rules, scanners, acquisition parameters, lesion contouring, image preprocessing, and machine learning techniques [20,21]. Furthermore, single-center datasets are almost always unavoidably small, increasing the risk of scarcely robust internal validation. Two papers on PI-RADS 3–5 recently showed that single-center models have a significant performance drop when applied to other centers’ data [22,23]. Efforts must therefore be made to (1) standardize as much as possible (as in radiomic features computation) [24]; (2) build large and multi-center datasets; (3) share developed models for external validation. This will allow us to understand whether general models can work even with center-specific variabilities or if center-specific models are needed instead.

On this basis, the aim of this work is manifold as follows: (a) to assess reproducible csPCa identification models found in the literature on an independent 80-patient dataset while providing details on their architectures; (b) to propose a new csPCa identification model for external validation based on robustly selected and easily obtainable radiomic and clinical features.

## 2. Materials and Methods

### 2.1. Study Population

We retrospectively retrieved medical and radiological data from our Institution’s Electronic Medical Records. According to urological indication, the initial population included 945 males who underwent prostate MRI (June 2016–March 2021) for suspected malignancy or active surveillance. From the original cohort, 706 patients were excluded for the following: (a) lack of one/more PI-RADS 3 lesion(s) as per PI-RADS v2.1 (*n* = 691); (b) no histopathological data within twelve months from MRI scan (*n* = 11); (c) poor image quality of diffusion-weighted (DWI) and/or in the T2-weighted sequences (*n* = 2) and apparent diffusion coefficient (ADC) map (*n* = 1). Accordingly, the final cohort included 80 males.

We collected the following clinical and laboratoristic data (Table 1): age, the most recent serological value of prostate-specific antigen (PSA; ng/mL), PSA density (total PSA/prostatic volume ratio), final histopathological analysis, and mean ADC value (mm^2^/s) calculated in a single 2D region of interest (ROI), i.e., the largest trackable circular area in the center of the lesion without exceeding the lesion margins.

### 2.2. MR Protocol and PI-RADS 3 Lesion Selection

Prostate MRIs were performed on a 3T scanner (Discovery MR750w GEM, GE Healthcare, Chicago, IL, USA), using a 16-channels pelvic anterior-array coil (GE Healthcare, Chicago, IL, USA), and with the patient supine. As per PI-RADS v2.1 criteria [3], MRIs were performed at least six weeks after any prostatic biopsy to avoid a possible source of diagnostic errors due to post-procedural bleeding foci. The standard MRI protocol is summarized in Table 2.

Blinded to pathological data, two radiology residents (A.C., P.N.F.; 3 years of experience) reviewed all MRIs in consensus, based on the current standard of care, considering the appearance of the lesions in the T2-w, DWI, ADC, and DCE sequences as per PI-RADS v2.1 [3]. For each patient, we selected a single target lesion (the largest one in case of multiple lesions). A board-certified radiologist (P.A.B.; 10 years of experience) validated the selection.

### 2.3. Pathological Examination

Each patient underwent a targeted biopsy of PI-RADS 3 lesions (4 cores) at our Institution. Biopsies were executed by a single operator with a total experience of more than 500 target fusion biopsies. We used the trans-rectal access and fusion technique with the reference MRI, a MyLabClassC ultrasound machine, and a virtual navigator fusion system (Esaote S.p.A., Genova, Italy) equipped with an end-fire endorectal probe. Additional systematic biopsies (12–16 cores, according to the following prostate volume: ≤60 mL vs. >60 mL) were performed (Figure 1 and Figure 2) [25]. It was thus possible to choose the prostate parenchymal tissue corresponding to the PI-RADS 3 target lesion as the reference standard. Gleason Score was assigned per 2005 ISUP recommendations (International Society of Urological Pathology) [26]. Each PCa-positive biopsy was evaluated according to the International Society of Urological Pathology 2014 consensus Gleason Grade Group system [11].

### 2.4. Lesion Segmentation

Anonymized DICOM files of FRFSE-T2-weighted sequences, DWI 2000 s/mm^2^ sequences, and ADC maps were exported and loaded on dedicated segmentation software, ITK-SNAP 3.8.0 (PICSL, University of Pennsylvania, Philadelphia, PA, USA) [27]. The 3D ROIs were manually delineated on every target lesion (Figure 3), both on T2-weighted sequences and DWI sequences/ADC maps in consensus by two radiology residents (A.C. and P.N.F.; 3 years of experience), and then validated by a board-certified radiologist (P.A.B.; 10 years of experience). Peripheral zone lesions were visible on both T2-weighted and DWI sequences/ADC maps. When a transitional zone lesion was not readily discernible on DWI/ADC maps, the segmentation area was delineated according to that traced on the T2-weighted sequence. An additional 3D ROI for each patient was outlined in the peripheral prostate zone to normalize intensity, avoiding potential focal lesions. Images were all corrected for magnetic field inhomogeneity (algorithm N4, 3D Slicer, http://www.slicer.org (accessed on 17 September 2022)).

### 2.5. Reproducible Literature Models Search and Assessment

We searched papers in the literature applying mpMRI radiomics as a tool to identify csPCa among PI-RADS 3 lesions. The following inclusion criteria were used: (1) PI-RADS 3 lesions identified according to PI-RADS v2.1 guidelines; (2) targeted biopsy as ground truth; (3) usage of IBSI-compliant tools for radiomic features computation; (4) adequate description of the methodological details (resampling grid, parameters in radiomic feature computation, selected radiomic features list, and model hyperparameters). Selected works’ details are reported in Table 3.

We extracted radiomic features from our 80-patient dataset using the work-specific processing and parameters for each work. We assessed the correlation between selected features and biopsy results through a univariate Mann–Whitney test applied to the entire patient sample. Then, pending the trained model availability, we retrained a model with the work-specific attributes and the work-specific input features on our 80-patient dataset, employing 100 repetitions of 5-fold stratified cross-validation and providing results in terms of sensitivity and specificity on the 500 validation sets. The following details are provided.

#### 2.5.1. T2-Based Hectors et al. Model

T2 images were normalized to range between mean ± 3σ (standard deviation) of the intensity in the volume of interest (VOI), resampled on a 0.5 × 0.5 × 0.5 mm^3^ voxel grid, and discretized to 64 bins. The selected 20 radiomic features were used as input in a scikit-learn (https://scikit-learn.org/stable/ (accessed on 17 September 2022)) Random Forest Classifier (maximum depth = 16; maximum number of features = none; minimum number of samples per leaf = 2; minimum number of samples required to split = 2; maximum number of leaf nodes = 16). A SMOTE oversampling of the minority class was adopted.

#### 2.5.2. T2 and DWI-Based Jin et al. Model

T2 and DWI image intensities were standardized; images were then resampled on a 1 × 1 × 1 mm^3^ voxel grid. Since the paper did not specify image discretization details, we used Pyradiomics default. The 4 selected radiomic features, along with patient age and PSA, were normalized using Z normalization and given input to a scikit-learn Logistic Regression classifier.

### 2.6. Proposal of a New Model to Be Validated by Other Centers

As clinical parameters, we assessed PSA, PSA density, age, and mean ADC value within a 2D ROI. Regarding radiomic features, we normalized T2 and ADC images by dividing voxel intensities by the average intensity computed within the corresponding normalization ROI. T2-weighted and ADC volumes were respectively resampled on a 0.4 × 0.4 × 3.0 mm^3^ and a 0.8 × 0.8 × 3.0 mm^3^ voxel grid through b-spline interpolation. In total, 958 radiomic features per patient were computed with Pyradiomics on original volumes (32 bin quantization) and HHH, LLL, HHL, and LLH coif1 wavelet decompositions (8 bin quantization for T2 and 16 bin quantization for ADC). Clinical and radiomic features more robustly related to GS were selected randomly dividing the 80 patients into 5 groups 100 times (maintaining the csPCa balance). In each of the 500 feature selection trials (4 groups at a time, 64 patients), the Mann–Whitney test assessed the univariate association between clinical/radiomic features and biopsy results. At the same time, we investigated the correlation between features using Spearman rank. The feature with the smallest univariate *p*-value was firstly selected. Then, features with increasing *p* values (if ≤0.01) were added only if characterized by an absolute value of the Spearman rank correlation <0.5 vs. already selected features. The final selected features pool contains features picked most times out of the 500 trials.

Univariate and multivariate models’ definitions and assessments were performed through 100 repetitions of a 5-fold stratified cross-validation scheme. Univariate models were defined by selecting thresholds maximizing the Youden index on training sets (4 groups, 64 patients) and assessed in terms of sensitivity and specificity on validation sets (1 group, 16 patients). The mean and standard deviation of sensitivity and specificity over the 500 validation trials were finally reported for each selected feature. For multivariate analysis, the following six classification models were considered: linear discriminant, linear, quadratic, and cubic support vector machine (SVM), classification tree, and K-nearest neighbours (KNN). All the possible feature combinations from the selected feature pool were assessed as classification model inputs. Models and optimal thresholds were identified on training sets and evaluated in terms of sensitivity and specificity on the corresponding 500 validation sets. Finally, a model to be shared for external validation was trained on the entire dataset. All the analysis was implemented in scikit-learn.

## 3. Results

Clinical-pathological results are described in Table 1. The detection rate for csPCa (Gleason score ≥ 3 + 4) at targeted and systematic biopsies in the case of PI-RADS 3, 4, and 5 was 32, 46, and 67%, respectively.

### 3.1. Assessment of Literature Features/Models

In Table 4, we reported the univariate association between radiomic features contained in Hectors’ [16] and Jin’s [19] models and biopsy results in our 80-patient dataset.

Regarding the performance of the re-implemented multivariate models relying on these features, Hector’s random forest model obtained a sensitivity of 40% ± 21% and a specificity of 71% ± 15%. Jin’s logistic regression model, which combined radiomic features, age, and PSA, obtained a sensitivity of 36% ± 20% and a specificity of 89% ± 10%.

### 3.2. Proposed Model

In the 500 feature selection trials, features more often selected and, therefore, more robustly correlated with biopsy were the following: (a) PSA Density (selection rate 100%); (b) a radiomic texture feature computed on the LLL wavelet band of T2-weighted images (T2-wavelet-LLL_glcm_InverseVariance, selection rate 87%); (c) a radiomic texture feature computed on the LLL wavelet band of ADC maps (ADC-wavelet-LLL_glszm_SizeZoneNonUniformity, selection rate 83%). The results of the univariate models’ assessment of the 500 validation trials are shown in Table 5. The correlation with histology was as follows: PSA density 66% ± 21% sensitivity and 71% ± 13% specificity; RF-T2 74% ± 21% sensitivity and 55% ± 15% specificity; RF-ADC 44% ± 19% sensitivity and 83% ± 13% specificity. In Table 5, the results obtained by the best multivariate model are also shown. The best multivariate model was a linear discriminant with the three features in input, which obtained a sensitivity of 80% ± 18% and a specificity of 76% ± 13% on the 500 test trials.

## 4. Discussion

Two works on mpMRI radiomics in prostate cancer recently showed that single-center models’ performance drops when models are applied to other center data [22,23]. This may be due to the too-small size of the training sample and to differences among centers in MR scanners, acquisition parameters, histological analysis, and segmentation. Protocol standardization, data, and model sharing will hopefully improve models’ reproducibility in the near future. Meanwhile, a step forward toward model generalizability assessment can be made as follows: (1) trying to test radiomics models proposed by others on an external dataset; (2) properly detailing radiomics works so that other groups can assess them on their own data. Unfortunately, to date, few groups in the literature have tested radiomic models developed by other centers. This is often due to the partial lack of details in radiomic papers, which prevents model re-implementation.

In this work, first, we tried to apply reproducible and standard-compliant literature research papers on mpMRI radiomics for PI-RADS 3 csPCa identification on our 80-patient dataset. We reviewed and summarized parameters, methodological choices, and results to simplify further validation by other groups. Then, we proposed a fully detailed and easily implementable new model for assessment on an external dataset. The following two works in literature satisfied our inclusion criteria: one from Hectors et al. [16], who proposed a T2-based model, and one from Jin et al. [19], who proposed a model relying on T2, DWI, age, and PSA. In total, 9 of the 20 radiomic features identified by Hectors et al. resulted significantly correlated to biopsy in our dataset (*p*-value ranging from 0.01 to 0.05), and 1 of the 4 radiomic features identified by Jin et al. resulted very significantly related to biopsy in our dataset (*p*-value 0.005). These features are all computed on T2 images, where peripheral and transitional zone lesion contours are easier to delineate and, therefore, likely less user-dependent.

In developing our radiomic model, we performed methodological choices that differed from the two groups. Mainly, we normalized intensities through a peripheral zone normalization ROI (as suggested by Bonekamp et al., 2018 [28]) and applied an FBN quantization with 32 bins on original images, 16 bins on ADC wavelet sub-bands, and 8 bins on T2 wavelet sub-bands. Other normalization/quantization schemes provided worse results and were not shown. The two radiomic features we found most robustly related to biopsy are both computed on the LLL wavelet sub-band, i.e., on a spatially smoothed version of T2 and ADC intensities inside the lesions.

The first feature, T2-wavelet-LLL_glcm_InverseVariance, reflects texture regularity. It was lower than 0.47 on csPCa, thus indicating that clinically significant tumors are characterized by a larger texture irregularity in the low-frequency sub-band of T2 images. This feature alone has good sensitivity (74%) but low specificity (55%). The second feature, ADC-wavelet-LLL_glszm_SizeZoneNonUniformity, measures the variability in the volumes of lesion zones (groups of connected voxels with similar intensity). It was larger than 17 on csPCa, thus indicating their more extensive zone size heterogeneity. This feature alone has a reasonable specificity (83%) but a low sensitivity (44%). It is worth noticing that the normalized version of this feature, computed on DWI, correlated to biopsy in the work of Jin. The lack of significance of Jin’s DWI feature on our dataset may be due to differences in the DWI acquisition protocol as follows: we used a b-value of 2000 mm/s^2^, while Jin used a b-value of 1500 mm/s^2^. We can therefore observe the following: (1) there is a coherence between our result and Jin’s; therefore, a greater zone size heterogeneity at the microstructural level is more likely related to malignancy; (2) the computation of this feature on ADC maps may be more robust and repeatable, being less dependent on DWI acquisition b-value.

Similarly to Jin et al., we then developed a multivariate model relying on radiomic and clinical features. However, we did not select clinical and radiomic features independently but included both within a single pool, to which a feature selection strategy was applied. Among clinical features, we selected PSA density, as alone might help tumor discrimination [29] (sensitivity of 66%, specificity of 71% in our dataset). A tri-variate model built on PSA density and the two readily available T2 and ADC radiomic features appears to discriminate csPCa with good confidence (sensitivity of 80%, specificity of 76%). We provided all the methodological details and are available to share trained models and optimized thresholds for external validation.

The model we are proposing is based on bi-parametric MRI (T2 and ADC sequences only, which are routinely acquired). It does not require time-consuming sequences, such as DCE images, which also expose patients to contrast medium-related possible side effects in an effort to build the simplest possible model able to identify csPCa, with the added benefit of reducing segmentation times and guaranteeing better standardization, thereby reducing the possible impact introduced by different DCE acquisition protocols and segmentation methods. Additionally, our model is based only on the following three features: the PSA density (routinely obtained in the standard workup of these patients) and the two radiomic features obtained in two standard mpMRI studies. We are aware that, in machine learning, wrapped and embedded feature selection methods that optimally combine a broader number of features within model optimization or even deep learning models, as seen in Bertelli et al. [30], are often used. However, we preferred to follow a different approach to try to obtain an explainable radiomic model, i.e., one able to explain lesion characteristics related to malignancy.

We think this approach has relevant implications in driving the adoption of radiomics in the clinical management of PI-RADS 3 lesions. From a radiological standpoint, it might complement the radiologist’s evaluation, increasing the overall diagnostic accuracy; on the other hand, from a clinical perspective, it might allow us to rule out unnecessary biopsies, avoiding the risk of procedure-related possible complications in selected patients.

This study has several limitations, mainly the small number of patients and the lack of an independent testing dataset. However, we tried to provide results as robustly as possible by performing both feature selection and model assessment in multiple subsamples. Furthermore, it is necessary to recognize that the pathological standard of reference should be the radical prostatectomy sample, not the histological result of the biopsy. In fact, our recent experience has shown that the combination of target and systematic biopsies fails to detect about 15% of the foci of csPCa at definitive pathology. However, only 4% turned out to be the index lesion (data not yet published). It could also be argued that the method used at our Institution for targeted biopsies is a rigid fusion, while there are elastic fusion technologies that can allow more accurate targeting. However, in a recent systematic review and meta-analysis, no significant difference in the detection of csPCa was identified when comparing rigid and elastic registration for MRI-TRUS fusion-guided biopsy [31].

Moreover, the high PI-RADS 3 lesion prevalence in the peripheral zone (82.5%) did not allow an investigation of any possible zone-related difference, and the overall sample size did not allow us to evaluate sector-related differences. We think there is a need for future research to assess not only regional differences between the transitional and peripheral zones but also different sectors’ related variability. Not clinically significant prostate cancers (Gleason score 3 + 3) were included in the negative group, as data in this last category were not conspicuous. However, this choice does not imply a particularly significant clinical limitation. MRI follow-up, with or without biopsy mapping, is usually performed for PI-RADS 3 lesions. Lastly, since the dataset was quite imbalanced (32.5% of tumors), we decided to optimize thresholds instead of using SMOTE since optimal thresholds provided better results.

## 5. Conclusions

Standard-compliant works with robust and detailed methodologies achieve comparable radiomic feature sets. Therefore, efforts to facilitate external validation of csPCa identification models with independent datasets are needed to help radiomics gain an effective role in the clinical workflow. In contrast, complex imaging models and protocols do not seem to be required. We showed indeed that PSA density, combined with two radiomic features computed on two routinely performed sequences (T2 and ADC), may potentially discriminate clinically significant prostate cancers (Gleason score ≥ 3 + 4).

## Figures and Tables

**Figure 1 jcm-11-06304-f001:**
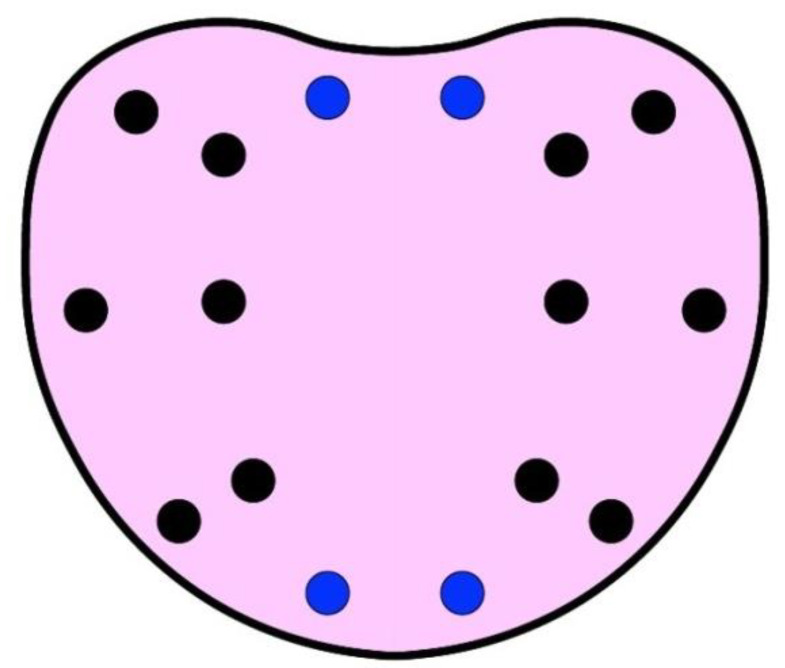
Scheme of systematic template for prostate biopsy. Black dots represent systematic biopsies. Blue dots represent additional systematic biopsies according to prostate volume (>60 mL).

**Figure 2 jcm-11-06304-f002:**
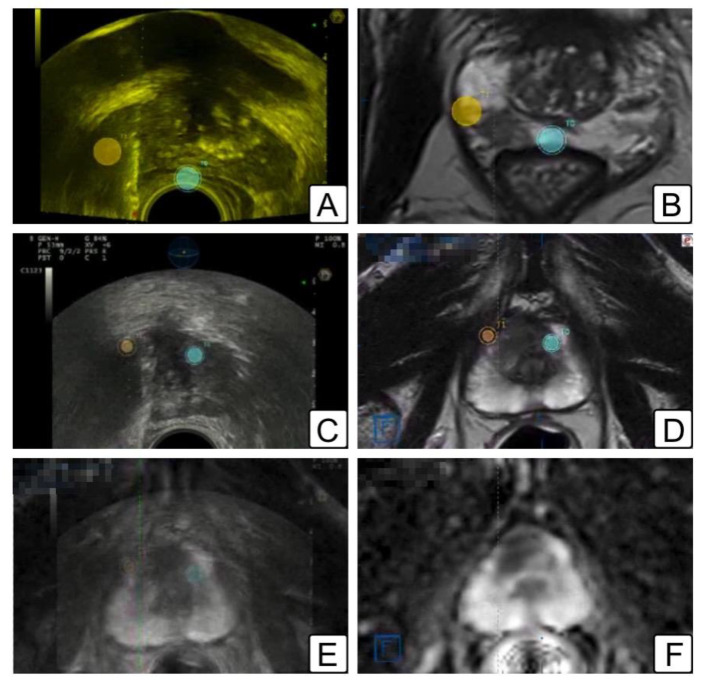
Illustrations of MRI/TRUS fusion biopsy. (**A**,**B**) Peripheral zone target biopsy: (**A**) trans-rectal ultrasound showing the location of the two lesions (orange and blue dots); (**B**) same lesions depicted in a T2-w MR (orange and blue dots). (**C**–**F**) anterior zone target biopsy: (**C**) trans-rectal ultrasound showing the location of the two lesions (orange and blue dots); (**D**) same lesions depicted in a T2-w MR (orange and blue dots); (**E**) fusion image overlapping T2-w MR image on top of transrectal ultrasound (lesions represented as orange and blue dots); (**F**) ADC map of the corresponding lesions.

**Figure 3 jcm-11-06304-f003:**
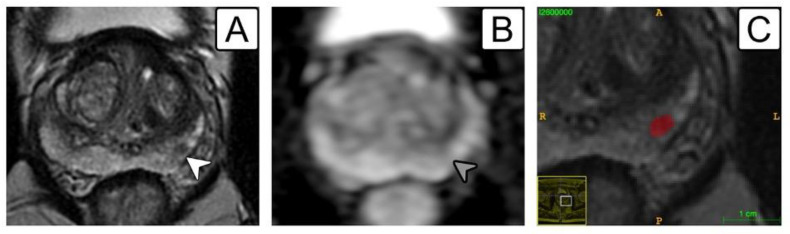
A 64-year-old patient with a PI-RADS 3 lesion in the left mid-gland peripheral zone. (**A**) Lesion on T2-w sequence depicted as a low-signal 5-mm nodule (white arrowhead); (**B**) same lesion highlighted on ADC map (grey arrowhead); (**C**) manual segmentation on ITK-SNAP (red label). Target biopsy revealed fibrosis with focal atrophy without evidence of prostate cancer.

**Table 1 jcm-11-06304-t001:** Characteristics of the final study population.

Population Data	
Total, *n*	80
Age (years), average ± SD (range)	65.2 ± 7.6 (45–81)
PSA (ng/mL), average ± SD (range)	6.8 ± 4.8 (0.5–29.6)
PSA Density, average ± SD (range)	0.15 ± 0.15 (0.01–1.09)
Mean ADC value within 2D ROI (mm^2^/s)	0.000825 ± 0.000253 (0.00026–0.00141)
PI-RADS 3 lesions histology, *n*/total (%)	
GS ≥ 3 + 4	26/80 (32.5%)
GS ≤ 3 + 3	16/80 (20.0%)
Negative, BPH, atrophy	38/80 (47.5%)
Site of PI-RADS 3 lesions, *n*/total (%)	
Transitional zone	14/80 (17.5%)
Peripheral zone	66/80 (82.5%)

PSA: prostate-specific antigen; PSA density is obtained by dividing PSA levels (ng/mL) by the volume of the prostatic gland (mL); PI-RADS: Prostate Imaging-Reporting and Data System; BPH: benign prostatic hyperplasia.

**Table 2 jcm-11-06304-t002:** MRI acquisition parameters.

	T1-w		T2-w		DWI
Acquisition plane	Axial	Axial	Axial, coronal, sagittal	Axial	Axial
Sequence	Fast spin-echo (SSFSE)	Gradient-recalled echo (GRE); before and after intravenous contrast (DCE)	Fast relaxation fast spin echo (FR-FSE)	Single-shot fast spin echo (SS-FSE)	b values: 50, 1000, 2000 s/mm^2^
Slice thickness	4 mm	3 mm	3 mm	4 mm	3 mm
Covered area	Pelvis	Prostate lodge and seminal vesicles	Prostate lodge and seminal vesicles	Pelvis	Prostate lodge

DWI: diffusion-weighted imaging; GRE: gradient-recalled echo; DCE: dynamic contrast enhancement; FRFSE: fast relaxation fast spin echo; SSFSE: single-shot fast spin echo.

**Table 3 jcm-11-06304-t003:** Selected models’ details.

Reference	Hectors 2021 [16]	Jin 2022 [19]
Number of subjects	240	103
Scanner	3T (GE Signa, Siemens Skyra)	3T (Siemens Skyra)
Endorectal coil	No	No
Radiomics MR sequences	T2	T2, ADC, DWI (1500 mm/s^2^)
ROIs	3D (1 operator)	3D (2 operators) on T2 (ADC/DWI registered to T2)
Radiomics platform	Pyradiomics	FeAture Explorer (Pyradiomics)
Intensity normalization	[μ − 3σ:μ + 3σ] inside the VOI	(x − μ)/σ
Resampling	0.5 × 0.5 × 0.5 mm^3^	1 × 1 × 1 mm^3^
Quantization	64 bins	Not specified
Model assessment	Cross-validation + independent test set	Independent test set
Selected radiomic feature details	Yes (20 features)	Yes (4 features)
Clinical parameters in the model	No	Yes (PSA, age)
Model	Random forest with SMOTE	Logistic regression
Radiomics model performances (test set)	AUC 0.76 Sensitivity 75.0% Specificity 79.6%	AUC 0.88 Sensitivity 83% Specificity 65%

**Table 4 jcm-11-06304-t004:** Univariate association between radiomic features contained in Hectors’s [16] and Jin’s [19] models and biopsy results in our 80-patient dataset; features with *p*-value ≤ 0.05 are in bold.

Hector’s Features	*p*-Value
T2-original_shape_Elongation	0.13
T2-original_shape_Flatness	0.14
T2-original_firstorder_10Percentile	0.94
T2-original_firstorder_InterquartileRange	0.40
T2-original_firstorder_Mean	0.43
T2-original_firstorder_Median	0.51
T2-original_firstorder_RootMeanSquared	0.38
T2-original_glcm_Autocorrelation	**0.01**
T2-original_glcm_DifferenceEntropy	0.06
T2-original_glcm_InverseVariance	**0.02**
T2-original_glcm_JointAverage	**0.01**
T2-original_glcm_JointEnergy	**0.04**
T2-original_gldm_LargeDependenceLowGrayLevelEmphasis	0.10
T2-original_glrlm_LongRunEmphasis	**0.05**
T2-original_glrlm_LongRunHighGrayLevelEmphasis	**0.01**
T2-original_glszm_GrayLevelVariance	0.12
T2-original_glszm_SizeZoneNonUniformity	**0.03**
T2-original_glszm_SmallAreaEmphasis	**0.01**
T2-original_ngtdm_Complexity	0.27
T2-original-ngtdm_Strength	**0.05**
**Jin’s Features**	***p*-Value**
T2-wavelet-HHL_glcm_ClusterTendency	**0.005**
DWI-original_glcm_ldmn	0.74
DWI-wavelet-LLL_glrlm_LongRunLowGrayLevelEmphasis	0.11
DWI-wavelet-LLL glszm_SizeZoneNonUniformityNormalized	0.75

**Table 5 jcm-11-06304-t005:** Selected features and performance of univariate and best multivariate models.

	Selection Rate	Sensitivity	Specificity
PSA Density	100%	66% ± 21%	71% ± 13%
T2-wavelet-LLL_glcm_InverseVariance	87%	74% ± 21%	55% ± 15%
ADC-wavelet-LLL_glszm_SizeZoneNonUniformity	83%	44% ± 19%	83% ± 13%
Trivariate linear discriminant model	-	80% ± 18%	76% ± 13%

## Data Availability

The data presented in this study are available on request from the corresponding author.

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
