# Peer review of "Radiomics in PI-RADS 3 Multiparametric MRI for Prostate Cancer Identification: Literature Models Re-Implementation and Proposal of a Clinical–Radiological Model"

_jcm, 2022, doi:10.3390/jcm11216304_

Round 1

Reviewer 1 Report

This is an interesting paper evaluating the role of radiomics in improving the diagnostic capacity of mpMRI with PIRADS 3 lesions.

This is a hot topic but also difficult in terms of reproducibility and bias.

Comments:

-        There are some misconfigurations in the pdf version that do not allow to properly revise all the paper. Please fix it.

-        There is an incongruency between the material and methods and table 4. It is described 20 features and only 19 are shown.

-        One of the main problems is the test consider as reference. Target biopsy has some limitation vs prostatectomy specimen as some false negative could occur. It should be included as limitations. The percentage of Significant prostate cancer is similar to that reported in literature, however, it would be helpful from my point of view to give the general results of target biopsy from the authors to see that is coherent with bibliography. Images provided of biopsy and targeting seem not very precise.

-        The other main problem is the use of biparametric sequence specifically in PIRADS 3 scores which is the scenario where DCE images have value in PIRADS classification, furthermore when most of the lesions are in the peripheral zone. This should be also included as limitation and discuss in the discussion section.

-        Have further sublocations of the lesions any association or influence in the prediction and radiomic ability (apex. Basal..)?

-        The percentage of peripheral zone in the table 1 and the discussion section varies. Please doublecheck.

Author Response

-     There are some misconfigurations in the pdf version that do not allow to properly revise all the paper. Please fix it.

ANSWER - We thank the reviewer for pointing out the issue. Unfortunately, we do not have control over PDF generation, which is done automatically. We revised the general formatting anyway, and everything should be good.

-     There is an incongruency between the material and methods and table 4. It is described 20 features and only 19 are shown.

      ANSWER - We thank the reviewer for this comment, which allowed us to spot a missing row in Table 4. We revised the Table and the manuscript accordingly.

-     One of the main problems is the test consider as reference. Target biopsy has some limitation vs prostatectomy specimen as some false negative could occur. It should be included as limitations. The percentage of Significant prostate cancer is similar to that reported in literature, however, it would be helpful from my point of view to give the general results of target biopsy from the authors to see that is coherent with bibliography. Images provided of biopsy and targeting seem not very precise.

ANSWER - We thank the reviewer for these important suggestions. First of all, we added in the Discussion the following limitation: “Furthermore, it is necessary to recognize that the pathological standard of reference should be the radical prostatectomy sample, not the histological result of the biopsy. In fact, our recent experience has shown that the combination of target and systematic biopsies fails to detect about 15% of foci of csPCa at definitive pathology. However, only 4% turned out to be the index lesion (data not yet published).”

As requested by the reviewer, we also added the detection rate of targeted and systematic biopsies in PIRADS 3,4, and 5. In the Results session, we added the following sentence: “The detection rate for clinically significant prostate cancer (Gleason score ³ 3+4) at targeted and systematic biopsies in case of PI-RADS 3, 4, and 5 was 32, 4,6, and 65%, respectively.”

Images provided of biopsy and targeting seem not very precise. We agree with the reviewer that a rigid fusion system could be considered less precise when compared to other fusion technologies. However, no significant difference in prostate cancer detection rate at targeted biopsy has been found in the literature. For this reason, we added in the Discussion the following sentence: “It could be argued that the method used at our Institution for targeted biopsies is a rigid fusion, while there are elastic fusion technologies that can allow more accurate targeting. However, in a recent systematic review and meta-analysis, no significant difference in the detection of clinically significant prostate cancerwas identified when comparing rigid and elastic registration for MRI-TRUS fusion-guided biopsy.”

-    The other main problem is the use of biparametric sequence specifically in PIRADS 3 scores which is the scenario where DCE images have value in PIRADS classification, furthermore when most of the lesions are in the peripheral zone. This should be also included as limitation and discuss in the discussion section.

ANSWER - We thank the reviewer for the remark, which allowed us to clarify that every PI-RADS score was assigned based on the current standard of care, considering the appearance of the lesions on the T2-w, DWI, ADC, and DCE sequences as per PI-RADS v2.1. The model we proposed did not include DCE images in an effort to build the simplest possible model able to identify csPCA, with the added benefit of reducing segmentation times and guaranteeing a better standardization, reducing the possible impact introduced by different DCE acquisition protocols and segmentation methods.

-     Have further sublocations of the lesions any association or influence in the prediction and radiomic ability (apex. Basal..)?

ANSWER - We thank the reviewer for the comment. Considering the overall sample size, regional differences have not been taken into consideration in the current study. We added it as a limitation, highlighting the need for future research not only in regional differences between transitional and peripheral zone but also in terms of sectors.

-     The percentage of peripheral zone in the table 1 and the discussion section varies. Please doublecheck.

      ANSWER - We thank the reviewer for the comment. We have revised the manuscript accordingly.

Reviewer 2 Report

The manuscript is a study concerning the role of radiomics analysis of MRI images of PI-RADS 3 prostatic lesions. The topic is certainly of interest to the readers, considering the increasing relevance of radiomics and artificial intellingence in medical images, but it needs some improvements.

First of all, taking into consideration the journal you are submitting to, it is not very clear whether the study is more "technical" or "clinical". At first reading, it seems the first one is true: for this reason, you should add some implications and consequences that your analysis may have in clinical practice and/or radiological evaluation.

Furthermore, you should better explain that MRI is the gold standard method for evaluation of prostate cancer itself, i.d. in the prostate gland, or even for the local evaluation, but not for a whole-body staging.

Line 55-56: "mpMRI radiomic analysis" is too generic, please add a brief sentence about the fields of application of radiomics in prostate cancer (e.g. lesion identification, prediction of biochemical recurrence, lymph node evaluation, etc.).

Again, when you declare the aim of your study, you should describe the context in which this analysis will take place. The same is true for the Conclusions.

Author Response

The manuscript is a study concerning the role of radiomics analysis of MRI images of PI-RADS 3 prostatic lesions. The topic is certainly of interest to the readers, considering the increasing relevance of radiomics and artificial intelligence in medical images, but it needs some improvements.

ANSWER - We thank the reviewer for the kind remark. We improved the overall quality of the manuscript.

First of all, taking into consideration the journal you are submitting to, it is not very clear whether the study is more "technical" or "clinical". At first reading, it seems the first one is true: for this reason, you should add some implications and consequences that your analysis may have in clinical practice and/or radiological evaluation.

ANSWER - We thank the reviewer for the comment, which allowed us to expand in the Discussion section on the clinical relevance it might have both in terms of clinical management and radiological evaluation.

Furthermore, you should better explain that MRI is the gold standard method for evaluation of prostate cancer itself, i.d. in the prostate gland, or even for the local evaluation, but not for a whole-body staging.

ANSWER - We thank the reviewer for the comment. We rephrased it, making it clearer that local staging is intended.

Line 55-56: "mpMRI radiomic analysis" is too generic, please add a brief sentence about the fields of application of radiomics in prostate cancer (e.g. lesion identification, prediction of biochemical recurrence, lymph node evaluation, etc.).

ANSWER - We thank the reviewer for the comment. We have revised the manuscript accordingly.

Again, when you declare the aim of your study, you should describe the context in which this analysis will take place. The same is true for the Conclusions.

ANSWER - We thank the reviewer for the comment. We have revised the manuscript accordingly.

Round 2

Reviewer 1 Report

all the comments have been addressed

Reviewer 2 Report

The manuscript has been improved and can be considered for publication in JCM.